

# Organic molecular tracers in the atmospheric aerosols from Lumbini, Nepal, in the northern Indo-Gangetic Plain: Influence of biomass burning

Xin Wan[1,9], Shichang Kang[2,8*], Quanlian Li[2], Dipesh Rupakheti[1,9], Qianggong Zhang[1,8], Junming Guo[1,9], Pengfei Chen[2], Lekhendra Tripathee[2], Maheswar Rupakheti[3,4], Arnico K. Panday[5], Wu Wang[6], Kimitaka Kawamura[7], Shaopeng Gao[1], Guangming Wu[1,9], Zhiyuan Cong[1,8*]

[1] Key Laboratory of Tibetan Environment Changes and Land Surface Processes, Institute of Tibetan Plateau Research, Chinese Academy of Sciences (CAS), Beijing 100101, China

[2] State Key Laboratory of Cryospheric Science, Northwest Institute of Eco-Environment and Resources, CAS, Lanzhou 730000, China

[3] Institute for Advanced Sustainability Studies (IASS), Potsdam 14467, Germany

[4] Himalayan Sustainability Institute (HIMSI), Kathmandu, Nepal

[5] International Centre for Integrated Mountain Development (ICIMOD), Kathmandu, Nepal

[6] School of Environmental and Chemical Engineering, Shanghai University, Shanghai 200444, China

[7] Chubu Institute for Advanced Studies, Chubu University, Kasugai 487-8501, Japan

[8] CAS Center for Excellence in Tibetan Plateau Earth Sciences, Beijing 100085, China

[9] University of Chinese Academy of Sciences, Beijing 100039, China

*Corresponding author:
E-mail address: shichang.kang@lzb.ac.cn; zhiyuancong@itpcas.ac.cn



## Abstract:

To better understand the characteristics of biomass burning in the northern Indo-Gangetic Plain (IGP), total suspended particles were collected in a rural site, Lumbini, Nepal during April 2013 to March 2014 and analyzed for the biomass burning tracers (i.e., levoglucosan, mannosan, vanillic acid, etc.). The annual average concentration of levoglucosan was $734 \pm 1043$ ng m$^{-3}$ with the maximum seasonal mean concentration during post-monsoon season ($2206 \pm 1753$ ng m$^{-3}$), followed by winter ($1161 \pm 1347$ ng m$^{-3}$), pre-monsoon ($771 \pm 524$ ng m$^{-3}$) and minimum concentration during monsoon season ($212 \pm 279$ ng m$^{-3}$). The other biomass burning tracers (mannosan, galactosan, p-hydroxybenzoic acid, vanillic acid, syringic acid, and dehydroabietic acid) also showed the similar seasonal variations. There were good correlations among levoglucosan, organic carbon (OC) and elemental carbon (EC), indicating significant impact of biomass burning activities on carbonaceous aerosol loading throughout the year in Lumbini area. According to the characteristic ratios: levoglucosan/mannosan (Lev/Man) and syringic acid/vanillic acid (Syr/Van), we deduced that the high abundances of biomass burning products during non-monsoon seasons were mainly caused by the burning of crop residues and hardwood while the softwood had less contribution. Based on the diagnostic tracer ratio (i.e., Lev/OC), the OC derived from biomass burning constituted large fraction of total OC, especially during post-monsoon season. By analyzing the MODIS fire spot product and five-day air-mass back trajectories, we further demonstrated that organic aerosol composition was not only related to the local agricultural activities and residential biomass usage, but was also impacted by the regional emissions. During the post-monsoon season, the emissions from rice residue burning in western India and eastern Pakistan could impact particulate air pollution in Lumbini and surrounding regions in southern Nepal. Therefore, our finding is meaningful and has a great importance for adopting the appropriate mitigation measures, not only at the local level but also by involving different regions and nations, to reduce the biomass burning emissions in the broader IGP region nations.



# 1. Introduction

Biomass burning such as domestic biofuel combustion, agricultural residues and wildfires contribute large amounts of pollutants in the atmosphere, which include trace gases (e.g., greenhouse gases like $CO_2$ and $CH_4$ and precursors of $O_3$) and aerosols, especially organic carbon (OC) and black carbon (BC) (Crutzen et al., 1979; Andreae and Merlet, 2001; van der Werf et al., 2006). These pollutants can cause adverse health effects, deteriorate air quality, affect earth's radiation budget and thus climate (Jacobson, 2014; Ramanathan and Carmichael, 2008). BC absorbs solar radiation and is primarily responsible for net positive radiative forcing (Bond et al., 2013). The assessment reports by the Intergovernmental Panel on Climate Change (IPCC) have reported that the radiative forcing (RF) of BC and OC from the biomass burning emissions can offset each other to give an estimated RF of +0.0 (-0.2 to +0.2) W m$^{-2}$, but there are substantial uncertainties because of the limited information about their sources, atmospheric loading, and composition of organic aerosols of which a significant fraction is also light absorbing organics, known as brown carbon (BrC) (IPCC, 2013). Due to the hygroscopic property of biomass burning emissions, they also have a pronounced indirect impact on climate by altering cloud microphysical properties (Andreae, 2009; Kawamura and Bikkina, 2016). In addition, biomass burning accounts for 4.4% of total carbon loss in terrestrial ecosystems and thereby plays an important role in the global carbon cycle (van der Werf et al., 2006; Hu et al., 2013). Thus biomass burning has drawn increasing global attention and concern in the recent decades.

To track the biomass burning emissions, organic molecular tracers such as anhydrosugars (Engling et al., 2009; Fu et al., 2012), resin acids (Kawamura et al., 2012; Fujii et al., 2015) and water soluble potassium (K$^+$) (Cheng et al., 2013; Sullivan et al., 2011; Urban et al., 2012) are widely exploited for their unique signatures. Levoglucosan (1,6-anhydro-β-D-glucopyranose), the most abundant component among anhydrosugars, is a distinct and most reliable tracer for biomass burning. It is formed by the pyrolysis of cellulose (Simoneit, 2002; Simoneit et al., 1999) and can remain stable in the atmosphere





without degradation for at least ten days (Fraser and Lakshmanan, 2000). Mannosan and galactosan
(isomers of levoglucosan), generated from the thermal decomposition of hemicellulose, can also be used
as biomass burning tracers (Simoneit et al., 1999; Sang et al., 2013; Urban et al., 2014). Additionally,
phenolic compounds such as p-hydroxybenzoic, vanillic and syringic acids are released to the
atmosphere during the combustion of lignin (Kawamura et al., 2012; Fu et al., 2012; Fujii et al., 2015).

Dehydroabietic acid is produced during the burning process of coniferous resins (Simoneit, 2002).
Furthermore, ratios of individual tracers can be also used as indicators for various biomass burning
types (Urban et al., 2012; Engling et al., 2009; Sang et al., 2013). For example, the levoglucosan to
mannosan ratios (Lev/Man) derived from softwood combustion are frequently lower than 10, but much
higher for the burning of hardwood and crop residues (Cheng et al., 2013). Potassium has also been

used as a conventional biomass burning tracer, but it may suffer from the interference from soil
re-suspension, sea salts and fire-works (Cheng et al., 2013; Urban et al., 2012).

The Indo-Gangetic Plain (IGP) (Fig. 1) in South Asia is one of the most densely populated and
polluted regions in the world. The large-scale urbanization, land use changes, industrial activities and
regional emission sources lead to high aerosol loadings over the entire IGP (Ram et al., 2010, 2012;

Lawrence and Lelieveld, 2010). It results in formation of widespread atmospheric brown clouds (ABCs)
in IGP and surrounding regions from the southern flank of the Himalayas to the northern Indian Ocean,
especially during long dry season from November to May every year (Ramanathan et al., 2005;
Bonasoni et al., 2010). A previous study conducted in India has demonstrated that residential biofuel
combustion and open burning are the largest sources of OC (87%) and BC (75%) emissions

(Venkataraman et al., 2005), much higher than fossil fuel combustion. Coincidentally, by using
radiocarbon, Gustafsson et al. (2009) confirmed that the biomass combustion accounted for two-thirds
of the bulk carbonaceous aerosols in India. Recently, several studies have demonstrated that the
atmospheric aerosols from biomass burning in the source region like IGP can be transported over long
distance to the Himalaya and Tibetan Plateau region (Cong et al., 2015a; Cong et al., 2015b; Luthi et al.,





2015; Kaspari et al., 2011; Li et al., 2016). After light-absorbing aerosols, particularly BC, get deposited on the snow and glacier surface, they accelerate melting of snow fields and glaciers (Xu et al., 2009). Some biomass burning tracers have been detected in the ice cores and snow samples from Tibetan Plateau (You et al., 2016; Gao et al., 2015).

Considering the serious air pollution in South Asia, the SusKat project (http://www.iass-potsdam.de/en/research/air-quality/suskat) was launched in May 2012 by the Institute

for Advanced Sustainability Studies (IASS), Germany with an aim of comprehensive understanding of air pollution (emission, atmospheric loading, physical/chemical processes, seasonal variation, their potential impacts) in northern South Asia, especially in Nepal, and identifying effective solutions that are rooted in solid science and carefully examined local conditions for reduction of air pollution impacts

in the region. IASS and the International Centre for Integrated Mountain Development (ICIMOD) jointly coordinated and carried out the SusKat-ABC international air pollution measurement campaign in Nepal during December 2012-June 2013 in collaboration with 16 other research institutes. Lumbini, a rural site in Nepal, which was one of the regional sites for the SusKat-ABC campaign, was chosen as a representative of the northern IGP region. The total suspended particles (TSP) sampling at Lumbini was

started during the SusKat-ABC campaign and continued, after the campaign, for a period of a year.

Biomass is the most common source of primary energy consumption in Nepal (WECS, 2010). In spite of the potential importance of emissions from regional biomass burning in the air quality, health, agriculture, glacier melting, and regional climate, the concentrations, chemical composition, and impact of biomass burning aerosols in Lumbini and broader surrounding regions, including Nepalese part of the

IGP and the foothills of the Himalaya have not been well characterized yet. Such studies are very critical to understand the transport mechanism of aerosols and air quality dynamics in the region. Therefore, in this study, we investigate for the first time the biomass burning tracers in the TSP in Lumbini, as a representative site in the northern edge of IGP, document their seasonal variations and evaluate the contributions of biomass burning to regional air quality. The characteristics of organic





aerosols revealed in this study may also be used as fingerprints to identify the source regions of air pollutants found in the remote Himalayas-Tibetan Plateau.

## 2. Methodology

### 2.1. Sampling site

Lumbini (Fig. 1, 27.49 ˚N, 83.28 ˚E, 100 m a.s.l.) is located in the Nepal's southern lowland plain

(also Terai region), termed as "bread basket of Nepal" due to the availability of very fertile land suitable for crop production. It is also worth mentioning that Lumbini, the birthplace of Buddha, is a UNESCO world heritage site (http://whc.unesco.org/). The high Himalayas are about 140 km north of Lumbini. The flat plains of southern Nepal and India surrounds the remaining three sides of Lumbini (Rupakheti et al., 2016). The sampling site is only about 8 km from the Nepal-India boarder in the south and within

the agricultural-residential setting. Paddy-wheat crop rotation system is the main planting pattern. The aerosol loading is very high at Lumbini, as also seen in data collected by ICIMOD and the Government of Nepal in 2016 (www.pollution.gov.np). A previous study reported that daily average $PM_{2.5}$ and $PM_{10}$ levels frequently exceeded the WHO guideline value (25 and 50 μg m$^{-3}$ for daily $PM_{2.5}$ and $PM_{10}$, respectively) during pre-monsoon season (Rupakheti et al., 2016). In terms of meteorological conditions,

Lumbini exhibits the typical characteristics of the IGP climate with wet monsoon season (June to September), dry winter season (December to February), dry-to-wet transition period or pre-monsoon season (March to May) and wet-to-dry transition period or post-monsoon season (October to November). The sampling experiment was performed on the roof of a tower (15 m above the ground) in the premises of the Lumbini International Research Institute (LIRI) within the Lumbini Master Plan

area.

### 2.2 Sample collection

From April 2013 to March 2014, the total suspended particles (TSP) samples were collected on a



weekly basis using a medium-volume sampler (KC-120H: Qingdao Laoshan Applied Technology Institute, Qingdao, China) at a calibrated airflow rate of 100 L min$^{-1}$. The sampling duration of each

sample was 24 hours (from 8:00 am local time to 8:00 am local time of the next day). The samples in May missed due to the equipment breakdown. Overall, 68 samples were collected on the quartz fiber filters (90-mm diameter; Whatman PLC, Maidstone, UK), which were prebaked at 550 ℃ for 6 h. Filters were weighed before and after sampling using a microbalance with a sensitivity of ±0.01 mg. They were equilibrated at constant temperature and humidity (25±3 ℃, 30±5%) for 72 hours before

and after sampling. The volume of air passing through each filter was converted into standard atmospheric conditions (25 ℃, 101.3 kPa). The samples were stored at -20 ℃ prior to laboratory analysis. To assess the potential contamination, field blank samples were also collected by placing the filters into the sampler with no air drawn.

## 2.3 Extraction, derivatization, and GC/MS determination

Biomass burning tracers including levoglucosan, mannosan, galactosan, p-hydroxybenzoic acid, vanillic acid, syringic acid, and dehydroabietic acid were detected using the methods adopted from Fu et al. (2008) and Wang et al. (2008). Briefly, filter aliquots (1.13–6.78 cm$^2$) were extracted with dichloromethane/methanol (v/v = 2:1) under ultrasonication for 30 minutes (20 ml×3). The solvent extracts were passed through quartz wool packed in a Pasteur pipette, concentrated by a rotary

evaporator under vacuum, and blown down to dryness with pure nitrogen gas. For sample derivatization, 50 μl of 99% N,O-bis-(trimethylsilyl)trifluoroacetamide with 1% trimethylsilyl chloride (BSTFA+1%TMCS) and pyridine (v/v = 2:1) was added to the dried extracts and then were reacted at 70 ℃ for 3 h.

    During the sample pretreatment procedure, the samples were spiked with appropriate amounts of

internal recovery standards, i.e., 2,000 ng methyl-β-D-xylanopyranoside (MXP, 99%, Sigma) and 200 ng deuterated ($D_3$) malic acid ($D_3$-malic acid; DMA, CDN isotopes, 99%). The derivatized fraction was



further dissolved to 200 µl with n-hexane and analyzed by a gas chromatography-mass spectrometry (GC-MS, Trace GC coupled to PolarisQ MSD, Thermo Scientific, USA). The GC instrument was equipped with a 30 m TG-5MS fused-silica capillary column (0.25 mm in inner diameter and 0.25 µm film thickness). Splitless injection of 1 µl sample was performed. The GC oven temperature program was initiated at 50 ℃, maintained for 2 min, a gradient of 30 ℃ min$^{-1}$ up to 120 ℃, then 6 ℃ min$^{-1}$ up to the final temperature of 300 ℃, maintained for 16 min. The MS was operated in the electron impact mode at 70 eV and an ion source temperature of 250 ℃. Full scan mode was used in the range of 50-650 Da. The total ion chromatogram of these tracers was presented in Fig. S1 in the supplement.

Recoveries for target compounds were better than 75% as obtained by spiking standards to pre-combusted quartz filters followed by extraction and derivatization. The internal standard recoveries obtained by the same method, better than 90% for MXP but much lower (70%) for $D_3$-malic acid. Field blank filters were analyzed by the procedure used by the samples above, but no target compound was detected. Duplicate analyses showed analytical errors were less than 15%. The recovery ratios of target compounds and internal standards are shown in Table S1. The method detection limits of the target compounds were 0.07-0.11 ng m$^{-3}$ at a total standard volume of 121 m$^{-3}$.

The following standards were obtained: levoglucosan (99%, Sigma-Aldrich), mannosan (98%, TRC), galactosan (97%, J&K), p-hydroxybenzoic acid (98%, AccuStandard), vanillic acid (98%, AccuStandard), syringic acid (99%, Sigma-Aldrich), dehydroabietic acid (97%, TRC). HPLC gradient grade reagents of dichloromethane (DCM), methanol (MEOH), n-hexane and pyridine for sample derivation were obtained from Jt Baker (USA). Individual standard stock solutions were prepared in MEOH at a concentration of 1,000 µg ml$^{-1}$. These composite standard solutions of three sugars, four organic acids and two internal standards were prepared by diluting individual standard stock solutions to 1 µg ml$^{-1}$ accurately.

**2.3 Determination of OC, EC and K$^+$**



OC and EC were analyzed based on the thermal/optical reflectance (TOR) method using the thermal/optical carbon aerosol analyzer (DRI Model 2001A, Desert Research Institute, USA), following the Interagency Monitoring of Protected Visual Environments (IMPROVE) protocol. The details on the method were described in Wan et al. (2015).

$K^+$, $Mg^{2+}$ and $Ca^{2+}$ were determined by using an ion chromatograph. In brief, an aliquot of filter (1.6 $cm^2$) was extracted with 10 ml ultrapure water (Millipore, 18.2 MΩ) with sonication for 30 min. The extracted solutions were filtrated with syringe-driven filters (MillexGV PVDF, 0.22 μm; Millipore, Ireland) to remove the impurities. Finally, $K^+$, $Mg^{2+}$ and $Ca^{2+}$ were determined using an ion chromatograph (Dionex ICS-320). The sample flow rate was 1.0 ml $min^{-1}$. The uncertainty was less than

5%. The detection limits were less than 0.01 μg $m^{-3}$ (Tripathee et al., 2016). The mass concentrations of OC, EC, $K^+$, $Mg^{2+}$ and $Ca^{2+}$ in this study were corrected from the field blank values.

### 2.4 Meteorological parameters

Daily average time series of various meteorological parameters during the sampling period are shown in Fig. 2. Wind speed (WS) was obtained with the sensor (model 05103-5, R.M. Young, USA) at

12 m above the ground (Rupakheti et al., 2016). Planetary boundary layer (PBL) data were obtained from the ECMWF (European Centre for Medium Range Weather Forecasts) database (www.ecmwf.int/en/forecasts). Ambient Temperature (T), pressure (P), relative humidity (RH), and visibility (V) data, which were reported for the Bhairahawa Airport, ca. 14 km to the east from Lumbini, were obtained from the website of Weather Underground (www.wunderground.com).

### 2.5 Backward trajectories and fire spots

In order to investigate the possible regional influence of air pollution outside of Lumbini, five-day backward air-mass trajectories starting at 500 m above ground level were calculated for every day at 00:00, 06:00, 12:00 and 18:00 UTC from April 2013 to March 2014 using the NOAA-HYSPLIT model



(version 4). The Global Data Assimilation System ($1°\times1°$) data from the National Center for Environmental Prediction (http://ready.arl.noaa.gov/gdas1.php) were used in this study. Cluster analyses were applied to elucidate the characteristic of air mass origins, in which three seed clusters were generated for each season. To illustrate the biomass burning activities in South Asia, the fire spots were obtained from Fire Information for Resource Management System (FIRMS) operated by the National Aeronautics and Space Administration (NASA) of the United States (https://earthdata.nasa.gov/earth-observation-data/near-real-time/firms).

## 3 Results and discussion

### 3.1 TSP, OC and EC

The summary of the TSP, OC and EC mass concentrations as well as biomass burning tracers observed at Lumbini are provided in Table 1. Their seasonal variations are shown in Fig. 3 while the temporal variations are shown in Fig. S2. The TSP mass concentrations at Lumbini ranged from 44.6 to 631 μg m$^{-3}$ with the annual mean of 196±132 μg m$^{-3}$ during the sampling period (April 2013 to March 2014). The TSP concentrations were high during post-monsoon (seasonal average: 354±150 μg m$^{-3}$), pre-monsoon (291±60.7 μg m$^{-3}$) and winter (260±112 μg m$^{-3}$) while much lower loadings were obtained during monsoon season (82.6±28.7 μg m$^{-3}$). The meteorological parameters indicated that the lower wind speed (Fig. 2c) and shallow planetary boundary layer (Fig. 2e) during the post-monsoon and winter could easily form the stagnant weather conditions, which is favorable for the accumulation of air pollutants. The visibility (Fig. 2f) was much lower in these seasons, confirming the poor air quality in the region. Meanwhile, the abundant precipitation during the monsoon led to the lower TSP concentrations through washout and scavenging.

The annual average mass concentrations of OC and EC in Lumbini were 32.8±21.5 μg m$^{-3}$ and 5.95±2.66 μg m$^{-3}$, accounting for 18.6±9.36% and 3.93±2.00% of TSP mass, respectively (Table 2). OC and EC exhibited a large seasonal variability during the sampling period (Table1, Fig. 3b and c). The



highest seasonally-averaged OC concentration (56.7±19.5 µg m$^{-3}$) occurred during the post-monsoon season and the lowest during the monsoon season (16.6±8.15 µg m$^{-3}$). Similar seasonal variations were also reported for other sites in the IGP, such as Delhi (Mandal et al., 2014) and Kanpur (Ram et al., 2012), i.e., maximum OC occurred in post-monsoon. However, it was different in Kathmandu, the capital of Nepal. Using a regional chemical transport models (WRF-STEM), Adhikary et al. (2007) reported the highest OC and EC concentrations in Kathmandu occurred in March and April (pre-monsoon season). For the seasonal variations of EC, it also clearly shared the similar seasonal pattern with OC. In addition, a strong correlation (R$^2$ = 0.67, P<0.001) was observed between OC and EC (Fig. 4a), indicating that they may be derived from common sources.

The OC/EC ratios in our study were relatively high, ranging from 2.41 to 10.3 with an average of 5.16. It has been reported that OC/EC ratios from biomass and biofuel burning emissions and secondary organic aerosols (SOA) are usually higher than those from fossil fuel sources (Cong et al., 2015a; Cao et al., 2013; Ram et al., 2012). Watson et al. (2001) have documented OC/EC ratios of 1.1 for motor vehicle emissions and 2.7 for coal combustion emissions. The high OC/EC ratios in Lumbini might be an indication of biomass combustion emissions. Similarly, Ram and Sarin (2010) have determined the OC/EC ratios of 7.87±2.4 in three urban sites (i.e., Allahabad, Kanpur and Hisar) in northern India, where were also under substantial impacts of biomass burning. Our finding of high OC/EC ratio at Lumbini in the northern edge of IGP indicates that it is a regional characteristic of the OC/EC ratio. As both OC and EC play important roles in radiative forcing and cloud microphysics and consequently on regional climate change and precipitation, knowledge of the OC/EC ratio for the IGP and surrounding regions is of particular significance for reducing uncertainties in quantification of the OC and EC regional radiative forcing.

## 3.2 Biomass burning tracers



Biopolymers including cellulose, hemicellulose, lignin, suberin, sporopollenin, chitin, etc. are essential components of biomass. When they are subjected to combustion, varieties of organic molecules are emitted to the atmosphere. Some of them can be used as specific tracers or markers of biomass burning sources, such as anhydrosugars, p-hydroxybenzoic acid, vanillic acid, dehydroabietic

acid and so on (Simoneit, 2002).

### 3.2.1. Anhydrosugars (pyrolysis products of cellulose and hemicellulose)

Anhydrosugars such as levoglucosan and its two isomers (mannosan and galactosan) are specifically formed during the pyrolysis of cellulose and hemicellulose and are widely used as tracers for biomass burning source (Simoneit et al., 1999; Engling et al., 2009; Sang et al., 2013; Ho et al.,

2014; Zhu et al., 2015; Zhang et al., 2015). In our ambient aerosols (i.e., TSP) over Lumbini, levoglucosan was found as the most abundant biomass burning tracer throughout the year (Table 1 and Fig. 3d), comprising 71.1% of the total biomass burning tracers detected (Fig. S3). The annual average concentration of levoglucosan in the aerosols was $734\pm1043$ ng m$^{-3}$, followed by mannosan ($33.2\pm32.2$ ng m$^{-3}$) and galactosan ($31.7\pm35.0$ ng m$^{-3}$) (Table1). The levoglucosan concentrations were at least one

order of magnitude higher than mannosan and galactosan. Levoglucosan represented $75.5\pm24.6\%$ of the total anhydrosugar concentration (sum of levoglucosan, mannosan and galactosan), while mannosan and galactosan only accounted for $12.3\pm12.9\%$ and $12.2\pm12.9\%$, respectively.

Since 90% of anhydrosugars exist in the particles with aerodynamic diameters less than 2 $\mu$m (Giannoni et al., 2012; Yttri et al., 2005), it is possible to compare levoglucosan concentrations in our

study with those reported in PM$_{2.5}$, PM$_{10}$ and TSP (Fig. 5). When compared with the areas that are significantly affected by biomass combustion such as Mt. Tai, China (391 ng m$^{-3}$) (Fu et al., 2008), Beijing, China (221 ng m$^{-3}$) (Yan et al., 2015), K-puszta, Hungary (309 ng m$^{-3}$) (Puxbaum et al., 2007), Gent, Belgium (477 ng m$^{-3}$) (Zdrahal et al., 2002), and Morogoro, Tanzania (253 ng m$^{-3}$) (Mkoma et al., 2013), the concentration levels of levoglucosan in Lumbini were relatively high (annual average:



734±1040 ng m$^{-3}$). The concentrations in Lumbini were even three orders of magnitude higher than those from the background site like Cape Hedo, Okinawa, Japan (3.09 ng m$^{-3}$) (Zhu et al., 2015). Therefore, Lumbini ranked among the highest biomass burning influenced sites in the world, whose level is comparable with New Delhi (1977 ng m$^{-3}$)(Li et al., 2014), Raipur (2180 ng m$^{-3}$) (Deshmukh et al., 2016), Rajim (2258 ng m$^{-3}$) (Nirmalkar et al., 2015) in India.

We further investigated the correlations of levoglucosan with OC, EC and K$^+$ (Fig. 6). Throughout the whole year, significant correlations were found for levoglucosan and OC (R$^2$=0.61, P<0.001), as well as levoglucosan and EC (R$^2$=0.42, P<0.001), suggesting their common sources from the combustion of biomass. It is noteworthy that no evident correlation was found between levoglucosan and K$^+$, hinting that there might be other sources of K$^+$ in Lumbini. The strong correlations between K$^+$
and Mg$^{2+}$ (R$^2$=0.84, P<0.001), K$^+$ and Ca$^{2+}$ (R$^2$=0.78, P<0.001) suggested that dust could be the major source of K$^+$ ( Fig. 4b and Fig. 4c).

To understand the relative contribution of levoglucosan to OC, the ratio of Lev/OC was calculated (Table 2). The maximum Lev/OC ratio was obtained during post-monsoon season with an average of 3.34±2.53%, followed by winter (2.40±2.10%), and pre-monsoon season (1.53±0.57%) while the
minimum value was obtained during monsoon with an average of 0.98±1.05% (Table 2). The Lev/OC ratio in the post-monsoon season was comparable with that of New Delhi (3.1±0.8%), where the carbonaceous aerosols were also attributed to biomass burning (Li et al., 2014). The ratio between levoglucosan and EC (Lev/EC) in Lumbini was also investigated, which showed the same descending order as Lev/OC ratio, i.e., post-monsoon > winter > pre-monsoon > monsoon season. Taken together,
the biomass burning emissions have a predominant influence on the aerosol composition in Lumbini, especially during post-monsoon and winter seasons. Based on the radiocarbon measurement ($^{14}$C) of carbon in the TSP, Li et al. (2016) found that the contribution of biomass burning to EC in Lumbini aerosols during these seasons were about 51.4%.



Mannosan and galactosan, which are mainly formed from the pyrolysis of hemicellulose, showed

the similar seasonal variations (Table 1, Fig. 3e and f) and significant correlations with levoglucosan (Fig. S5). Previous studies have used the mass concentration ratio of levoglucosan to mannosan (Lev/Man) to differentiate the type of biomass burning. Based on the previous biomass burning studies, Cheng et al. (2013) showed that high Lev/Man ratio from hardwood burning is more than 10 whereas it is less than 10 from softwood combustion. Engling et al. (2009) reported higher ratios (more than 10)

for emissions from burning hardwood and crop residues. Much higher Lev/Man ratios of more than 40 were obtained from chamber experiment burning of rice straw, wheat straw and maize stalks (Zhang et al., 2007; Engling et al., 2009). In this study, the seasonal average Lev/Man ratios were 19.7±2.58 (15.0–23.6), 27.2±17.2 (0.76–45.7) and 16.1±13.1 (0.33–31.4) during pre-monsoon, post-monsoon and winter, respectively (Table 2 and Fig. S4a). The ratios from our study during these non-monsoon

seasons were close to the reported ratios of Lev/Man for hardwood and crop residues. Interestingly, dramatically high Lev/Man ratios of 44.3 and 45.7 were observed on 10th and 13th November (during the post-monsoon season), which were more likely associated with the crop residue combustion (Zhang et al., 2007; Engling et al., 2009).

During the monsoon season (Fig. S4a), Lev/Man ratios varied over a wide range (0.42–22.0),

which may be associated with the photochemical degradation of levoglucosan under the conditions of high solar radiation and higher humidity. Hoffmann et al (2010) have proposed that degradation fluxes of levoglucosan by OH radicals is higher in wet season than in dry seasons. Another possible reason is due to the different biomass types burning such as the burning of softwood and hardwood. Nevertheless, the mechanism is unclear yet in Lumbini and more detailed studies are needed in the future.

### 3.2.2 p-Hydroxybenzoic, vanillic and syringic acids (lignin pyrolysis products)

The phenolic compounds such as p-hydroxybenzoic acid, vanillic acid, and syringic acid are major tracers for burning of lignin and have been used as specific tracers for different biomass types (Simoneit





et al., 1993; Simoneit, 2002). Specifically, p-hydroxybenzoic acid is indicative of emission from combustion of herbaceous plants (e.g. grass and crop). On the other hand previous studies have reported

that vanillic acid is mainly emitted from burning softwood (gymnosperms) with less contribution from burning hardwood and grasses whereas syringic acid is dominantly produced from burning hardwood (angiosperms) and grasses (Simoneit, 2002; Myers-Pigg et al., 2016).

These three phenolic compounds were detected in all of the TSP samples at Lumbini, although their contents were about one order of magnitude lower than levoglucosan (Table 1). To our knowledge,

this study presents for the first time the phenolic compounds in aerosols collected in Nepal. The concentrations of p-hydroxybenzoic acid ranged from 0.23 to 39.7 ng m$^{-3}$ in the entire sampling with an average of 9.36±10.8 ng m$^{-3}$ (Table 1). Vanillic and syringic acids had the annual mean values of 7.59±8.87 ng m$^{-3}$ and 5.81±6.02 ng m$^{-3}$, respectively, which are at a similar level to p-hydroxybenzoic acid. The seasonal variations of these organic acids were coincident with levoglucosan (Fig. 3), and they

exhibited significant correlations with levoglucosan (Fig. 7a, levoglucosan and p-hydroxybenzoic acid ($R^2$=0.64, P<0.001); Fig. 7b, levoglucosan and vanillic acid ($R^2$=0.85, P<0.001); Fig. 7c, levoglucosan and syringic acid ($R^2$=0.81, P<0.001)). This implies that these four tracers are emitted from the common sources i.e. biomass combustion. These biomass burning tracers and diagnostic ratios suggest that the burning of herbaceous plants (grass, dung, agricultural waste, etc.), hardwood and softwood made a

mixed contribution to the organic aerosols in Lumbini.

Recently, the mass ratio of syringic acid to vanillic acid (Syr/Van) has been suggested as an indicator to further distinguish the relative importance of different vegetation burned (Myers-Pigg et al., 2016). According to the previous studies, the Syr/Van ratios for burning woody angiosperm (hardwood) and non-woody angiosperm vary from 0.1 to 2.44, while it is much lower (i.e. 0.01-0.24) for burning

gymnosperm (softwood) (Shakya et al., 2011; Myers-Pigg et al., 2016). Regarding the ambient aerosol samples of Lumbini, Syr/Van ratios varied in the range from 0.39 to 2.58 with an average of 0.94±0.46 during the sampling period (Table 2), suggesting that hardwood and grass (including crop residue) are




more likely sources for the biomass burning aerosols in Lumbini. This finding is in agreement with the results derived from anhydrosugar pyrolysis products (Lev/Man), as discussed in Section 3.2.1.

### 3.2.3 Dehydroabietic acid (pyrolysis product of conifer resin)

Dehydroabietic acid is produced by direct emission from the pyrolytic dehydration of resins that are present in the bark surfaces, needle leaves and the woody tissues of conifers (Simoneit et al., 1993). Its emission may be different from those of burning lignin and cellulose (Simoneit et al., 1993). Hence, dehydroabietic acid can be used as a more specific tracer for the burning of conifer trees (softwood). In this study, concentrations of dehydroabietic acid fluctuated from 0.60 to 4.85 ng m$^{-3}$ with annual average concentration of 1.75 ng m$^{-3}$ (Table 1). It was about two orders of magnitude lower than that of levoglucosan, and one order of magnitude lower than lignin pyrolysis products (i.e., p-hydroxybenzoic, vanillic and syringic acids). Obviously, dehydroabietic acid showed the lowest levels among the biomass burning tracers, which demonstrated that the burning of conifer trees in Lumbini was scarce. Stockwell et al. (2016) also reported that hardwoods (such as Sal (*shorea robusta*), Bakaino (*melia azedarach*), etc.) are widely used in residential cooling and heating activities in the Lumbini area. In Nepal, especially in rural areas, biomass burning is still a major domestic energy source for cooking and heating. Moreover, open field burning of agriculture residues (specifically, wheat and paddy straws) is a common way to clean up the croplands after harvesting.

### 3.3 Contribution of biomass burning to OC

As mentioned above, anhydrosugars (levoglucosan, mannosan and galactosan) from the pyrolysis of cellulose and hemicellulose can be considered as indicators of biomass burning emission. In this study, anhydrosugars account for 1.79±1.66% of OC annually, with the highest seasonally averaged value in post-monsoon (6.67%) which is a major crop residue burning season in the IGP region. The contribution of anhydrosugars to OC in Lumbini is comparable with that of the Amazon rainforest sites





(2%−7%) where the burning of forests happened intensively (Graham et al., 2002). These values were higher than those of found in the Pearl River Delta sites in China (0.59%−3.12%) that were directly affected by biomass combustion in southern China (Ho et al., 2014).

Since levoglucosan is the most abundant anhydrosugar, the ratio of levoglucosan to OC measured in source samples of biomass burning has been widely used to quantitatively estimate the biomass burning contribution to OC (Zdrahal et al., 2002; Puxbaum et al., 2007; Wang et al., 2007a; Zhang et al., 2010, 2012; Sang et al., 2011). Andreae and Merlet (2001) have reported that the Lev/OC ratios ranged from 8.0% to 8.2 % (average of 8.14%) in the burning of savanna, grassland, tropical and extratropical forests, biofuel and agricultural residues. Sullivan et al. (2008) reported that levoglucosan in OC was 7.6% for rice straw burning. Moreover, Zhang et al. (2007) reported an average of 8.27% (with a range of 5.4%−11.8%) of levoglucosan in OC during the burning of cereal straw (corn, wheat, and rice). The contributions of biomass burning to OC (based on enrichment factor reception modeling approach) can be inferred as follows:

$$Contributions\ of\ biomass\ burning\ to\ OC\ (\%) = \frac{\left(\frac{[lev]}{[OC]}\right) ambient}{\left(\frac{[lev]}{[OC]}\right) source} \times 100\%$$

Although the Lev/OC ratios in the biomass burning source emissions vary among different types of biomass fuels and burning conditions (e.g., Mochida et al., 2010), the average value of 8.14% has been commonly chosen to estimate biomass burning (BB) contributions to OC (i.e. BB-OC) (Fu et al., 2014; Mkoma et al., 2013; Sang et al., 2011; Ho et al., 2014). In this study, we also use the ratio of 8.14%. Table 2 shows the seasonal contributions of BB-OC to OC. Fig. 8 presents monthly (a) and temporal (b) variations of contributions of BB-OC to OC. The calculation showed that BB-OC contributed 19.8±19.4% (ranging between 0.05% and 81.9%) of OC in Lumbini aerosols on an annual basis. This is higher than



the contribution of BB-OC in the Pearl River Delta in China (13.1 %) (Ho et al., 2014), Hong Kong (6.5%−11%) (Sang et al., 2011). Moreover, maximum contributions of BB-OC to OC in our study were observed during post-monsoon (41.0±31.1%), which was as high as 58.7±21.7% in November.

These results indicated that biomass burning in Lumbini, especially in post-monsoon season, significantly contributes to ambient organic aerosols and can significantly affect the regional air quality. Intriguingly, in Godavari, a rural site at the southeastern edge of the Kathmandu Valley in Nepal, Stone et al. (2010) reported that primary biomass burning sources contributed 21±2% to OC in $PM_{2.5}$ mass concentrations. It should be noted that our estimation was based on the empirical values. Therefore, to restrict the uncertainty of this estimation, the direct determination of those critical ratios from major emission sources (local and regional) are needed in the future. Moreover, the contribution of SOA formation from biomass burning (BB-SOA) should also be taken into consideration, since BB-SOA such as some phenol compounds and methoxyphenols can account for a significant fraction of particulate matter derived from wood combustion, accounting for 21% and 45% by total aerosol masses (Hawthorne et al., 1989; Yee et al., 2013; Zhang et al., 2015).

### 3.4 Sources and regional transport

Besides local emissions, the regional transport of air pollutants also appears to have an impact on the Lumbini region. To better understand the source areas, we analyzed five-day air mass backward trajectories using HYSPLIT model along with fire spots acquired with MODIS during different seasons, and the results are shown in Fig. 9. Five-day was chosen because the atmospheric residence times of carbonaceous aerosols are about a week (Reddy and Boucher, 2004). It should be noted that, the fire spots of MODIS could reliably represent the occurrence and distribution of active open biomass burning, like the burning of crop residue in the farmland or the forest fires. However, residential biomass burning for cooking and heating cannot be detected by the satellite observation.

During the pre-monsoon season, the influence of biomass burning on carbonaceous aerosols





observed in Lumbini is high as seen in Table 2, Fig. 3 and Fig. 5, but it is somewhat less than in post-monsoon and winter. The concentration levels of OC and EC in pre-monsoon are about 3 times higher than those in the monsoon season. According to the MODIS active fire observation in this period

(Fig. 9a), there were substantial active fire spots detected in the area surrounding Lumbini, especially in the areas to the east and south of Lumbini, which is due to the burning of crop residues (mainly wheat) by the local farmers after the harvest (Ram and Sarin, 2010). Although the air mass trajectories originated in Pakistan and India in the west and moved eastward towards the site (Lumbini), few fire spots were detected along the trajectories of the air mass parcels while passing over eastern Pakistan

and western India. Fire spots are seen to be more concentrated in areas in India and Nepal surrounding Lumbini (Fig. 9a). Therefore, the high burden of biomass burning aerosols found in Lumbini in pre-monsoon season could be attributed to the local agricultural burning.

During the monsoon season, there were fewer fire spots detected by MODIS (Fig. 9b). During the summer, the arrival of southwest monsoon brings moisture from the Arabian Sea and Bay of Bengal,

leading to frequent and heavy precipitation events and thereby causing the wet season (June to September) in South Asia. Therefore, the biomass burning emission was observed to be the least in the monsoon season, which were reflected by not only in the concentrations of OC, EC and biomass burning tracers, but also in the ratios like Lev/OC.

For the seasonality of the composition of aerosols in Lumbini, the most striking feature drawn in

this study is that the air quality (TSP and its chemical composition) of post-monsoon season received the most significant influence from biomass burning, especially in November (Fig. 3, 5 and 8). The aerosol loadings, i.e., TSP, OC, EC, levoglucosan, Lev/OC and BB-OC/OC ratios, all together pointed out the importance of biomass burning emissions to atmospheric aerosols in Lumbini region. As shown in Fig. 9c, intensive fire spots (likely farm fires) were observed during post-monsoon in northwest India

and eastern Pakistan (i.e., Punjab) while fire spots were scarce around Lumbini. The burning of rice straw residues in Punjab has been well documented previously (Singh and Kaskaoutis, 2014; Jain et al.,



2014). Extensive agricultural burning in this area lasts for more than three weeks during every post-monsoon season, causing the smoke widely spread as a blanket nearly over the whole IGP (Fig. S6), with very high $O_3$ and CO loadings (Kumar et al., 2016) and particulate matter concentrations (Rastogi et al., 2016). Concerning the situation in Lumbini, the flow of air mass is dominated by westerlies in post-monsoon season. Fifty-three percent of the air mass trajectories to Lumbini originated around the most polluted northwestern India and eastern Pakistan (Fig. 9c). Therefore, it could be reasonably deduced that the serious biomass burning emissions from agricultural practice in this area could be transported over long distance to Lumbini in Nepal. The corresponding satellite image from MODIS (Terra) also showed the intensive air pollution plumes that flow toward the east over the IGP, reaching the Bay of Bengal (Fig. S6).

Relatively few fire spots were observed during winter around Lumbini (Fig. 9d), but the contribution of biomass burning was the second highest in the whole year (details were presented in Section 3.1 and 3.2). This phenomenon may be caused by the large amounts of indoor burning of mainly biomass fuel as well as small but numerous fires outside the houses to keep warm from the cold during intensive cold waves and winter fogs that shrouds much of IGP, including Lumbini, every winter, which could not be detected (especially indoor fires) by the satellites. In addition, the majority of the air mass backward trajectories to Lumbini during this season were local. The temperature during this season is the lowest of the whole year (Fig. 2a), therefore, local biomass burning for household heating has enhanced. In addition, the weak wind speed and low boundary layer in winter were conducive to trap the air pollutants near the ground in Lumbini.

## 4. Summary and environmental implication

Organic carbon, elemental carbon, and biomass burning tracers (levoglucosan, mannosan, galactosan, p-hydroxybenzoic acid, vanillic acid, syringic acid and dehydroabietic acid) were studied in the aerosols collected at Lumbini in the northern edge of the Indo-Gangetic Plains (IGP). We analyzed



their abundances, seasonal variations, and possible sources. We found that levoglucosan was the predominant biomass burning tracer among the measured biomass burning emission tracers, which showed a clear seasonal cycle with the post-monsoon maximum and monsoon minimum. Levoglucosan showed significant correlations with OC and EC, highlighting the biomass burning as a significant

contributor to the particulate air pollution in Lumbini. High levoglucosan/mannosan and syringic acid/vanillic acid ratios were observed during non-monsoon seasons, indicating that the main burning materials were mixed crop residues and hardwood with a minor contribution of softwood. Based on a diagnostic tracer ratio (i.e., levoglucosan/OC), the OC derived from biomass burning constitutes a large fraction of total OC in ambient aerosols, accounting for nearly 20% on annual average and as high as 40%

in the post-monsoon season.

Besides the chemical composition of aerosols, the fire spots observed by MODIS and air mass backward trajectories further suggested that the sources of biomass burning aerosols in Lumbini were dynamic in different seasons. In the pre-monsoon season, a high burden of biomass burning aerosols appeared to be due to the burning of wheat residues by the local farmers in the region. While in the

summer monsoon season, it exhibited the least influence of biomass burning. The peak loading of biomass burning aerosols in the post-monsoon was most likely due to long distance transport of emissions from agro-residue burning regions in the northwestern India and eastern Pakistan (e.g., Punjab). In winter, the local usage of biofuel for domestic heating may contribute to concentrations of organic aerosols under the favorable meteorological conditions (i.e., shallow planetary boundary layer

and calm winds).

Through the comprehensive analysis of aerosol composition, this study demonstrated that the biomass burning plays an important role in atmospheric carbonaceous aerosols and air quality in Lumbini and surrounding regions in the northern IGP. Given the adverse effects of biomass burning aerosols on air quality, public health, sensitive ecosystems, and regional climate, our study indicates

need for (i) simultaneous investigation of characteristics of carbonaceous aerosols at multiple site in



relatively poorly studied regions in northern IGP, the Himalayans foothills, and the remote sites in the Himalayas and Tibetan Plateau, which is critical for understanding transport of air pollutants from South Asia to Tibetan, and their impacts; (ii) adaptation of appropriate mitigation measures to reduce emissions of particulate and gaseous air pollutants, notably from biomass burning. Besides changing

agricultural practices, switching to clear fuels or to more advanced cook stoves that burn the biofuels for cooking and heating more completely and efficiently will be needed to reduce emissions. Our work clearly revealed that air pollution observed at Lumbini has both local and regional origin. Therefore, local actions to reduce air pollution in South Asia are essential but not sufficient because reduction of regional emissions requires involvement of different regions and nations. As Lumbini is a World

Heritage Site of universal value as the birthplace of Buddha, reduction of air pollution at this important site requires local, regional and global attention.

Recently, the light absorbing organic carbon, i.e., brown carbon (BrC) in the aerosols has been a frontline research topic because brown carbon is reported to act in the climate system as a warming factor like black carbon. Biomass burning has been suggested to be the predominant source of brown

carbon. Our understanding of organic carbon and brown carbon in this region is far from adequate and hence large uncertainties remain in quantifying their radiative forcing. Therefore, considering the strong influence of biomass burning to atmospheric OC and BrC over the IGP and surrounding regions, especially regions to its north, the OC, EC and BrC in this region deserves more research in the future.

**Acknowledgments**

We gratefully thank the staff at the Lumbini site for the sample collections, and all of the individuals and groups participating in the SusKat project. The authors acknowledge C. Cüppers and M. Pahlke of the Lumbini International Research Institute (LIRI) for providing the space and power to run the instruments at the LIRI premises. Authors also acknowledge Bhoj Raj Bhatta for the support in operation of the site. This study was supported by the Strategic Priority Research Program (B) of the



Chinese Academy of Sciences (XDB03030504), and the National Natural Science Foundation of China (41522103, 41225002, and 41630754).

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



**Table 1** Summary of the mass concentration of TSP, OC, EC ($\mu g\ m^{-3}$), and biomass burning tracers (ng $m^{-3}$), and OC/EC ratio in Lumbini across different seasons during April 2013 to March 2014.

| Compounds | Annual (n=68) Mean±SD | Range | Pre-monsoon (n=18) Mean±SD | Range | Monsoon (n=32) Mean±SD | Range | Post-monsoon (n=8) Mean±SD | Range | Winter (n=10) Mean±SD | Range |
|---|---|---|---|---|---|---|---|---|---|---|
| TSP ($\mu g\ m^{-3}$) | 196±132 | 44.6-631 | 291±60.7 | 210-440 | 82.6±28.7 | 44.6-143 | 354±150 | 114-631 | 260±112 | 120.78-451 |
| OC ($\mu g\ m^{-3}$) | 32.8±21.5 | 5.78-81.6 | 45.9±17.8 | 21.3-81.6 | 16.6±8.15 | 5.78-37.4 | 56.7±19.5 | 18.9-76.2 | 42.2±20.6 | 8.29-65.8 |
| EC ($\mu g\ m^{-3}$) | 5.95±2.70 | 1.91-12.9 | 7.39±1.85 | 4.79-10.9 | 4.04±1.30 | 1.91-7.50 | 10.00±2.66 | 4.72-12.9 | 6.24±1.64 | 2.24-8.86 |
| OC/EC ratio | 5.16±2.09 | 2.41-10.3 | 6.39±2.38 | 3.00-10.3 | 3.96±1.05 | 2.41-7.05 | 5.50±0.87 | 4.01-6.43 | 6.54±2.68 | 3.48-10.2 |
| Levoglucosan | 734±1043 | 1.31-5083 | 771±524 | 162-1829 | 212±279 | 1.31-889 | 2206±1753 | 6.03-5083 | 1161±347 | 3.96-3181 |
| Mannosan | 33.2±32.2 | 0.73-124 | 38.5±24.8 | 7.16-89.3 | 16.4±13.9 | 0.73-45.7 | 63.8±37.4 | 7.89-111 | 52.9±50 | 1.32-124 |
| Galactosan | 31.7±35.0 | 1.24-142 | 34.5±25.1 | 6.38-86.8 | 14.2±12.9 | 1.24-45.1 | 66.5±48.6 | 4.21-133 | 55±52.3 | 3.47-142 |
| p-Hydroxybenzoic acid | 9.36±10.8 | 0.23-39.7 | 16.6±10.5 | 3.95-39.7 | 2.05±2.07 | 0.23-7.08 | 20.9±13.5 | 0.92-33.1 | 10.5±9.55 | 0.93-26.3 |
| Vanillic acid | 7.59±8.87 | 0.28-37.1 | 10.2±6.89 | 2.57-25.8 | 2.26±2.06 | 0.28-7.85 | 17.5±11.4 | 0.62-33.9 | 12.0±12.5 | 1.29-37.1 |
| Syringic acid | 5.81±6.02 | 0.20-25.0 | 6.81±4.66 | 1.42-17.6 | 3.06±3.03 | 0.20-10.6 | 12.1±8.20 | 0.42-22.8 | 7.72±8.72 | 0.86-25 |
| Dehydroabietic acid | 1.75±0.96 | 0.60-4.85 | 2.13±1.00 | 0.64-3.85 | 1.17±0.37 | 0.60-1.97 | 3.00±0.71 | 1.82-3.82 | 1.94±1.12 | 0.65-4.85 |
| Anhydrosugars | 799±1105 | 3.53-5327 | 843±574 | 175-2005 | 243±302 | 3.53-972 | 2336±1834 | 18.1-5327 | 1269±1447 | 8.74-3447 |
| Total lignin and resin products | 24.5±25.5 | 1.58-93.3 | 35.7±22.4 | 8.88-86.9 | 8.54±7.11 | 1.58-25.2 | 53.6±31.3 | 3.78-93.3 | 32.2±31.2 | 5.02-90.8 |
| Total biomass burning tracers | 824±1128 | 5.32-5421 | 879±596 | 184-2091 | 252±309 | 5.32-995 | 2390±1861 | 21.9-5421 | 1301±1477 | 13.8-3538 |




**Table 2** Annual and seasonal means of Lev/Man and Syr/Van ratios and contributions of OC and EC to TSP, of Lev to OC, EC and TSP, of total anhydrosugars to OC and TSP, of total BB tracers to OC and TSP, and of BB-OC to OC and TSP.

| Contributions | Annual (n=68) Mean±SD | Range | Pre-monsoon (n=18) Mean±SD | Range | Monsoon (n=32) Mean±SD | Range | Post-monsoon (n=8) Mean±SD | Range | Winter (n=10) Mean±SD | Range |
|---|---|---|---|---|---|---|---|---|---|---|
| OC/TSP (%) | 18.6±9.36 | 6.86-62.1 | 15.9±5.69 | 7.76-25.1 | 21.5±12.0 | 7.66-62.1 | 16.7±3.88 | 10.3-21.9 | 16.0±5.43 | 6.86-26.6 |
| EC/TSP (%) | 3.93±2.00 | 1.60-10.4 | 2.61±0.77 | 1.64-4.95 | 5.29±2.06 | 2.50-10.4 | 3.09±0.77 | 1.60-4.13 | 2.64±0.95 | 1.82-4.32 |
| Lev/OC (%) | 1.61±1.58 | 0.004-6.67 | 1.53±0.57 | 0.72-2.87 | 0.98±1.05 | 0.00-3.57 | 3.34±2.53 | 0.03-6.67 | 2.40±2.10 | 0.02-4.83 |
| Lev/EC (%) | 9.85±11.9 | 0.03-47.9 | 10.5±6.55 | 2.86-23.2 | 4.36±5.02 | 0.03-17.7 | 19.9±15.7 | 0.13-41.7 | 18.3±20.3 | 0.07-47.9 |
| Lev/TSP (%) | 0.31±0.33 | 0.003-1.30 | 0.27±0.18 | 0.06-0.68 | 0.23±0.28 | 0.00-0.99 | 0.58±0.48 | 0.01-1.30 | 0.42±0.45 | 0.00-1.13 |
| Anhydrosugars/OC (%) | 1.79±1.66 | 0.01-6.99 | 1.68±0.62 | 0.81-3.14 | 1.16±1.12 | 0.01-3.92 | 3.55±2.63 | 0.10-6.99 | 2.63±2.26 | 0.04-5.24 |
| Anhydrosugars/TSP (%) | 0.34±0.35 | 0.01-1.36 | 0.29±0.20 | 0.06-0.75 | 0.26±0.30 | 0.01-1.08 | 0.62±0.51 | 0.02-1.36 | 0.46±0.48 | 0.01-1.23 |
| Total BB/OC (%) | 1.86±1.69 | 0.01-7.11 | 1.75±0.64 | 0.85-3.24 | 1.21±1.14 | 0.01-4.03 | 3.63±2.67 | 0.12-7.11 | 2.70±2.31 | 0.07-5.37 |
| Total BB/TSP (%) | 0.35±0.36 | 0.01-1.38 | 0.31±0.20 | 0.07-0.77 | 0.27±0.30 | 0.01-1.11 | 0.64±0.52 | 0.02-1.38 | 0.47±0.49 | 0.01-1.25 |
| Lev/Man | 15.1±11.2 | 0.33-45.7 | 19.7±2.58 | 15.0-23.6 | 9.19±7.99 | 0.42-22.0 | 27.2±17.2 | 0.76-45.7 | 16.1±13.1 | 0.33-31.4 |
| Syr/Van | 0.94±0.46 | 0.39-2.58 | 0.67±0.15 | 0.43-0.93 | 1.27±0.47 | 0.48-2.58 | 0.67±0.08 | 0.56-0.82 | 0.58±0.12 | 0.39-0.74 |
| BB/OC (%) | 9.02±12.8 | 0.02-62.4 | 9.47±6.44 | 1.99-22.5 | 2.61±3.43 | 0.02-10.9 | 27.1±21.5 | 0.07-62.4 | 14.3±16.5 | 0.05-39.1 |
| BB-OC/OC (%) | 19.8±19.4 | 0.05-81.9 | 18.8±6.95 | 8.85-35.2 | 12.0±12.8 | 0.05-43.9 | 41.0±31.1 | 0.39-81.9 | 29.5±26.1 | 0.22-59.4 |
| BB-OC/TSP (%) | 3.79±4.06 | 0.03-15.9 | 3.29±2.20 | 0.69-8.41 | 2.80±3.40 | 0.03-12.2 | 7.17±5.95 | 0.06-15.9 | 5.13±5.49 | 0.04-13.8 |





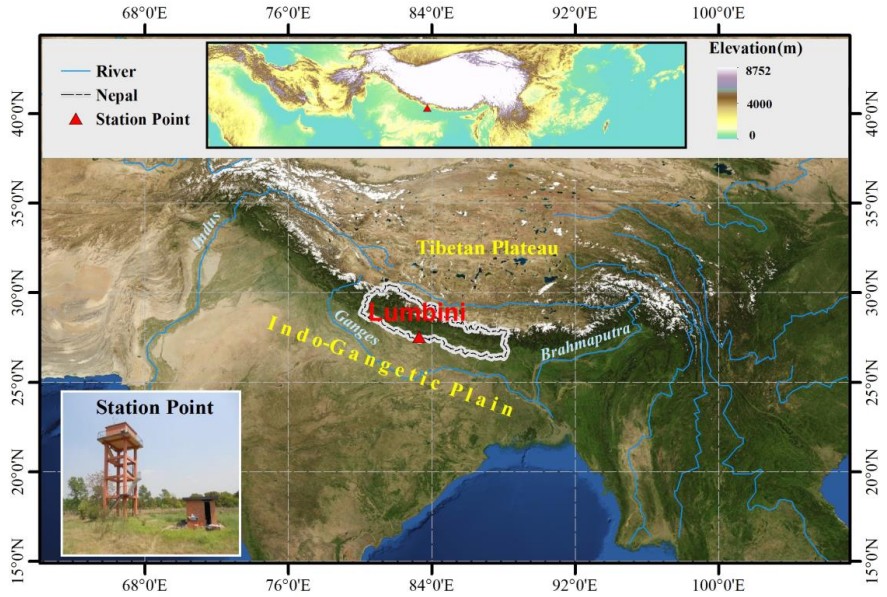

**Fig. 1.** Map showing the location and general setting of the sampling site of Lumbini in the northern edge of the Indo-Gangetic Plain.






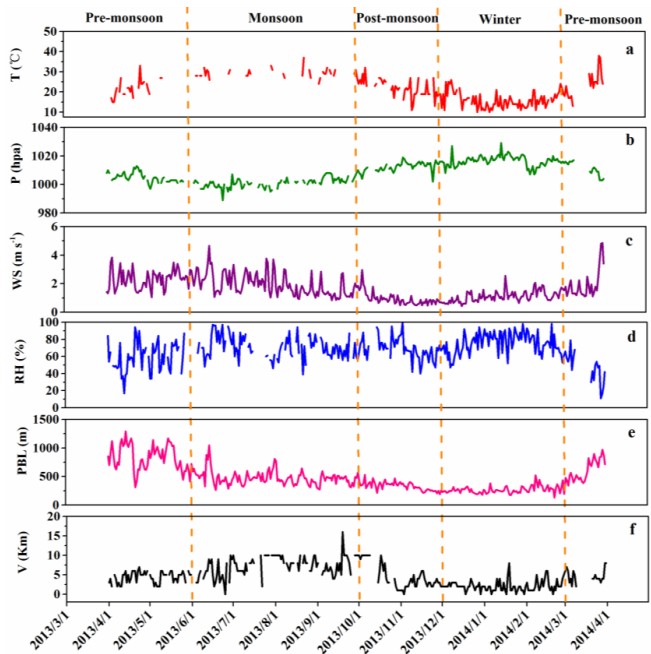

**Fig. 2.** Time series of ambient temperature (T), atmospheric pressure (P), wind speed (WS), relative humidity (RH), planetary boundary layer (PBL), and visibility (V) at Lumbini from April 2013 to March 2014.




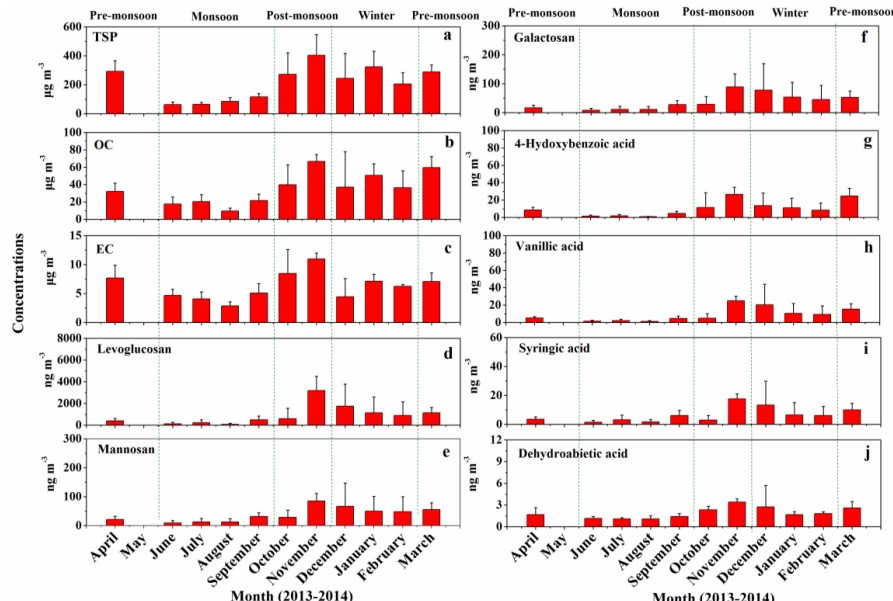


**Fig. 3.** Seasonal variations in concentrations of TSP, OC, EC, and organic tracers in aerosols collected in Lumbini from April 2013 to March 2014 (The data of May, 2013 was missing due to the equipment breakdown).



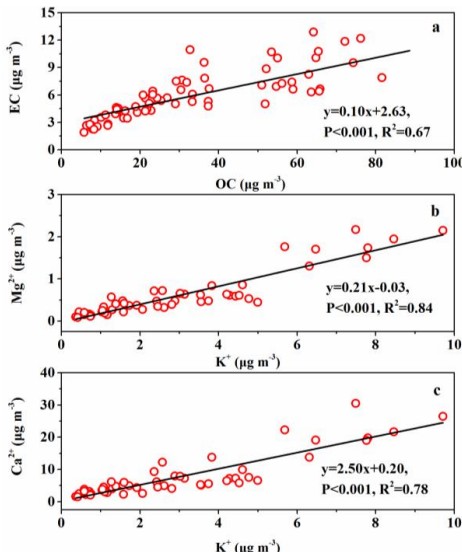

**Fig. 4.** Correlations between (a) OC and EC, (b) $K^+$ and $Mg^{2+}$, (c) $K^+$ and $Ca^{2+}$ during

the entire sampling period of a year (April 2013 to March 2014).





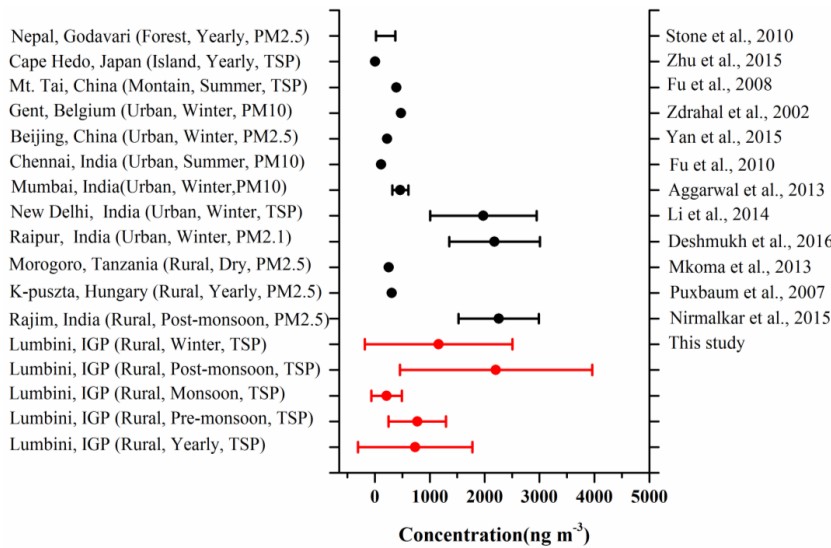

**Fig. 5.** Comparison of levoglucosan concentrations in Lumbini with other sites in different locations worldwide.





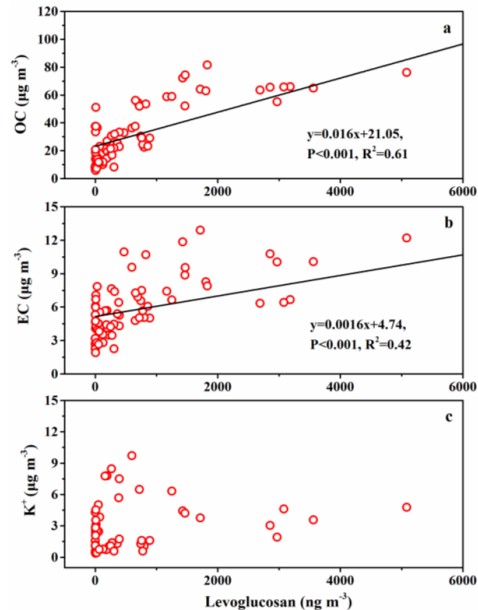

**Fig. 6.** Correlations between (a) levoglucosan and OC, (b) levoglucosan and EC, and (c) levoglucosan and $K^+$ during the whole year.

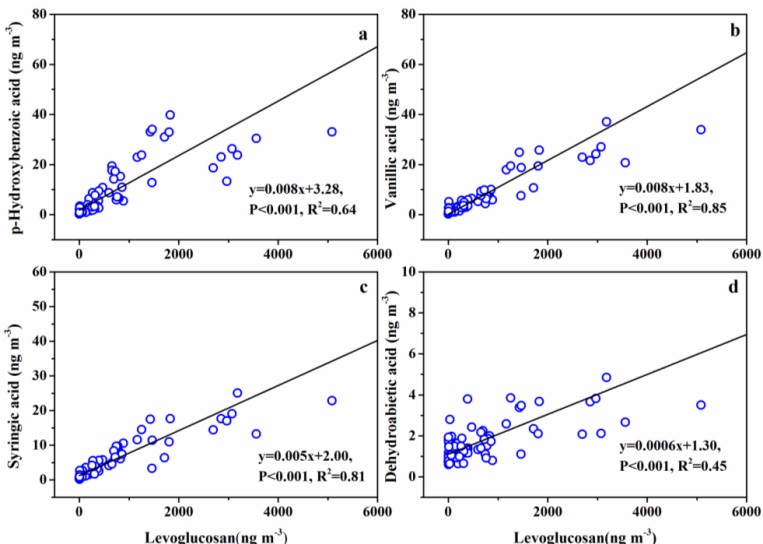

**Fig. 7.** Correlations between (a) levoglucosan and p-hydroxybenzoic acid, (b) levoglucosan and vanillic acid, (c) levoglucosan and syringic acid, (d) levoglucosan and dehydroabietic acid in the Lumbini aerosols during the whole year.




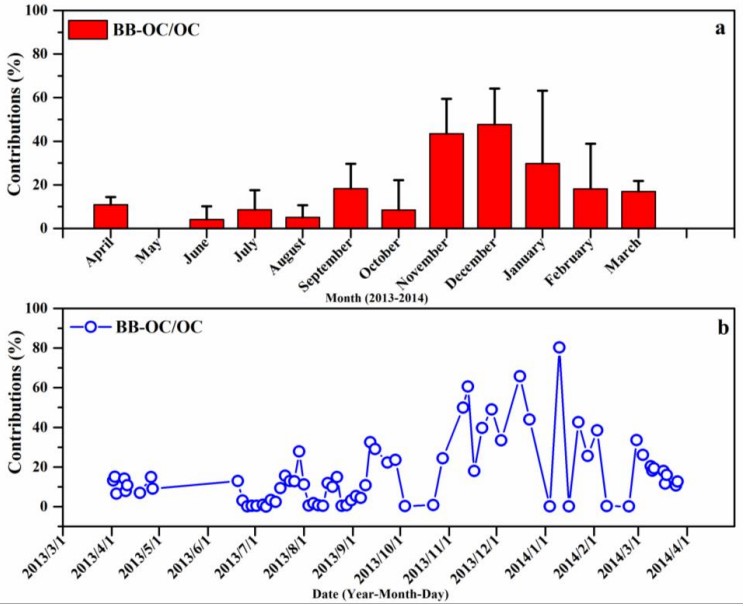

**Fig. 8.** (a) Monthly and (b) temporal variations in the contributions of biomass burning organic carbon (BB-OC) to organic carbon (OC) calculated from the diagnostic tracer ratio of Lev/OC (The data of May, 2013 was missing due to the equipment breakdown).




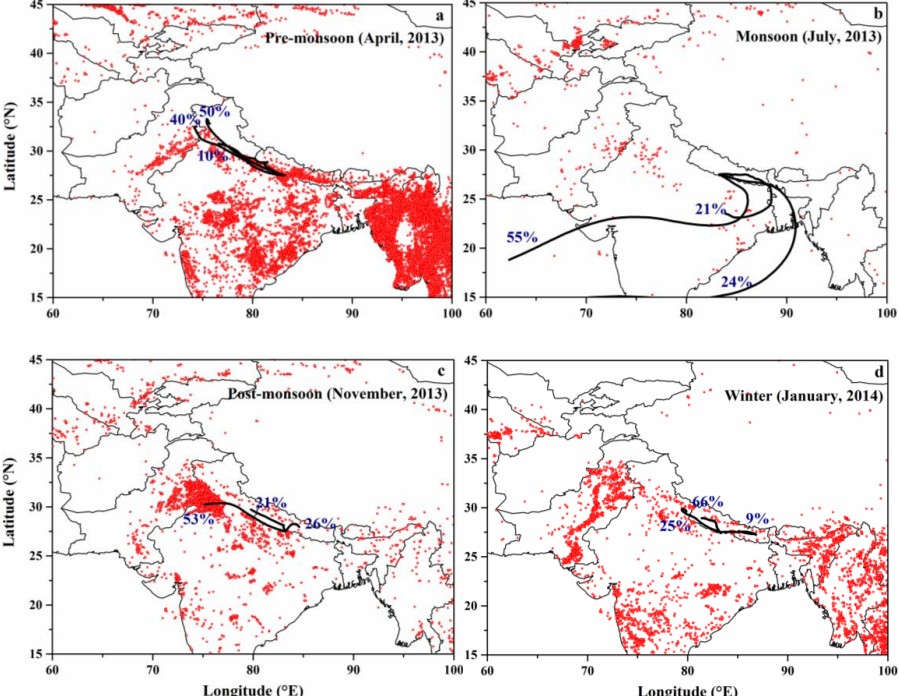

**Fig. 9.** Spatial distributions of fire spots observed by MODIS and air mass origins to Lumbini, the North IGP shown by clusters of 5-day backward trajectories arriving at 500 m above ground level during April 2013–March 2014. The numbers in each panel indicate the percentages of daily trajectories with the origins.