# Peer review of "Organic molecular tracers in the atmospheric aerosols from Lumbini, Nepal, in the northern Indo-Gangetic Plain: Influence of biomass burning"

_Atmospheric Chemistry and Physics, 2016_

## Referee Comment (RC1) · Anonymous Referee #1 · 31 Mar 2017

General comments

This work reports measurement data of a few biomass burning organic tracers in TSP samples collected from Lumbini, Nepal over a year. With the data, the authors investigated the influence of biomass burning from both local emissions and regional transport on the atmospheric aerosol under different meteorological conditions (pre-monsoon, wet monsoon season, post-monsoon and dry winter season). Contributions of biomass burning to OC mass were also estimated using levoglucosan data. By selecting three kinds of representative biomass burning tracers (anhydrosugars, lignin pyrolysis products and dehydroabietic acid), the authors also qualitatively evaluated the relative importance of different biomass fuel types in this region. Although the results are more or less expected, the data set is a useful addition to the database of ambient PM chemical characteristics in SE Asia, a region where such data is restively scarce. I list below specific comments for authors to consider in their revision.

Specific comments

1. Were anions analyzed? (Cations were analyzed, as indicated in the experimental part). If anion data is available, please include them in Table 1 and also comment on sulfate concentration abundance and their seasonal variation characteristics, as sulfate might provide insights into extent of regional influence. If no anion data is available, the authors could take a look at $NH_4+$ data, which was expected to be analyzed by IC along with other cations ($Ca_2+$, $Mg_2+$, $K+$).

2. In the paragraph starting at Line 247, OC/EC ratio is discussed. The authors need to be more cautious in comparing the OC/EC ratio from their TSP samples to those in the literature, which are largely data associated with PM2.5 samples. OC on the coarse particles has significant contributions from non-combustion sources (such as vegetative detritus, dust). It has been reported OC/EC ratio in the coarse mode was much larger than those in the accumulation mode (e.g. Yu et at, ACP, 10, 5107–5119, 2010), due to the significant presence of non-combustion OC. It would be misleading without commenting on the influence of non-combustion primary OC on the OC/EC ratio.

3. In the paragraph starting at Line 324, the authors discussed the variation range of Lev/Man ratio and attributed the wide rage observed (0.42-22.0) to photochemical degradation of levoglucosan. Such a reason is unlikely, as Mannosan also degrades and its degradation rate is likely similar to that of levoglucosan, considering their similar chemical structures.

4. Lines 130-131: "The aerosol loading is very high at Lumbini,.." Please provide more quantitative information (e.g., annual average or typical seasonal average concentrations).
5. line 143: The sampling schedule is on a weekly basis. Was it a regular schedule (i.e., one sample every 7 days or a random day in a week)?

6. line 152: Please provide more information on the field blank samples: how frequently was field blank filters collected?

7. Line 180: 121 m3: does this correspond to the air volume passing through the entire filter? Why it was not 144 m3 (0.100 m3/min x 60x24)?
* * *

---

## Referee Comment (RC2) · Anonymous Referee #2 · 22 Apr 2017

Wan et al reports a study of organic aerosols in Lumbini, a site of northern Indo-Gangetic Plain (IGP) with the emphases on the influence of biomass burning. They found that organic aerosols in Lumbini was not only related to the local agricultural activities and residential biomass usage, but also impacted by the regional emission. Although some previous work has reported organic aerosols in this region (mostly in India), this study does add valuable knowledge for the scientific community to better understand the abundance, composition and sources of organic aerosols in north IGP. Such information is useful for adapting proper mitigation measures, and thereby reducing the air pollution in South Asia. Generally, this paper is well written and logically ordered. The statement is convincing since it is based on a comprehensive dataset

and appropriate discussion. Therefore, I recommend this paper to be published after a revision.

Specific comments: 1. The English writing needs to be improved. Some sentences and description are redundant. For example, the sentence of Lines 53-58 is too long, which is hard to follow. 2. Line 177, do you have any idea about the low recovery of malic acid? 3. Line 190. K+ should be replaced by major ions, because in the following section, Ca2+ and Ma2+ were also included in the IC analysis. 4. Line 200, as stated by the authors, the concentrations of OC, EC and major ions reported in this work were corrected by their field blank values. So, the levels of such constituents in the field blank filters should be presented. 5. Lines 248-259, in general, relatively high OC/EC ratio represents a high contribution of secondary organic aerosol (SOC) as mentioned in many publications. According to your current dataset of OC, EC as well as the tracers, is it possible to estimate the importance of secondary organic aerosols? Considering the emissions and meteorological conditions in this region, SOC maybe also very important, but now little is discussed inside the manuscript. 6. Line 470, due to the low PBL height in the winter, could you provide its average value, instead of the current subjective description? 7. Line 505, clear or clean? 8. I suggested the authors to move the Fig S6, regarding the MODIS image of the regional emission and transport of smoke plume in the fall, to the main text. 9. As highlighted by the authors, the BrC in the organic aerosols deserve more research in the future. Actually, in the paper by Stockwell et al (2016, ACP), some optical properties of BrC in biomass-burning aerosols have already been determined. So, such new progress should be reflected in this manuscript. 10. In Figure 5: Fu et al. (2010) was missed in the Section of References.

Reference: Stockwell, C. E., Christian, T. J., Goetz, J. D., Jayarathne, T., Bhave, P. V., Praveen, P. S., Adhikari, S., Maharjan, R., DeCarlo, P. F., Stone, E. A., Saikawa, E., Blake, D. R., Simpson, I. J., Yokelson, R. J., and Panday, A. K.: Nepal Ambient Monitoring and Source Testing Experiment (NAMaSTE): emissions of trace gases and lightabsorbing carbon from wood and dung cooking fires, garbage and crop residue burning, brick kilns, and other sources, Atmospheric Chemistry and Physics, 16, 11043-11081, 10.5194/acp-16-11043-2016, 2016.

---

## Author Comment (AC1) · 28 Apr 2017

**Response to referee #1**

We would like to thank the referee for the interest in our work and the helpful comments and suggestions to improve our manuscript. We have carefully considered all comments and the replies are listed below.

**General comments**

This work reports measurement data of a few biomass burning organic tracers in TSP samples collected from Lumbini, Nepal over a year. With the data, the authors investigated the influence of biomass burning from both local emissions and regional transport on the atmospheric aerosol under different meteorological conditions (pre-monsoon, wet monsoon season, post-monsoon and dry winter season). Contributions of biomass burning to OC mass were also estimated using levoglucosan data. By selecting three kinds of representative biomass burning tracers (anhydrosugars, lignin pyrolysis products and dehydroabietic acid), the authors also qualitatively evaluated the relative importance of different biomass fuel types in this region. Although the results are more or less expected, the data set is a useful addition to the database of ambient PM chemical characteristics in SE Asia, a region where such data is restively scarce. I list below specific comments for authors to consider in their revision.

**Specific comments:**

1. Were anions analyzed? (Cations were analyzed, as indicated in the experimental part). If anion data is available, please include them in Table 1 and also comment on sulfate concentration abundance and their seasonal variation characteristics, as sulfate might provide insights into extent of regional influence. If no anion data is available, the authors could take a look at $NH_4^+$ data, which was expected to be analyzed by IC along with other cations ($Ca^{2+}$, $Mg^{2+}$, $K^+$).

**Response:** Yes, the major ions ($SO_4^{2-}$, $NO_3^-$, $Cl^-$, $NH_4^+$, $Ca^{2+}$, $Mg^{2+}$, $K^+$, and $Na^+$) were detected in our study. However, considering the theme of this work focuses on the organic aerosols, especially the influence of biomass burning, the data of major ions were not fully presented and discussed. They will be reported in other manuscript by one of the co-authors. Therefore, we only discuss $Ca^{2+}$, $Mg^{2+}$ and $K^+$ here.

The annual average concentration of $SO_4^{2-}$ in Lumbini aerosols was $6.85 \pm 5.65$ µg m$^{-3}$, accounting for $21.0\% \pm 6.93\%$ of the total major ions ($SO_4^{2-}$, $NO_3^-$, $Cl^-$, $NH_4^+$, $Ca^{2+}$, $Mg^{2+}$, $K^+$, and $Na^+$). The concentration was similar with the studies in Agra (5.9 µg m$^{-3}$ in TSP) and Durg (8.88 µg m$^{-3}$ in PM10), India (Satsangi et al., 2013; Deshmukh et al., 2012) while lower than that in Beijing, China (13.6 µg m$^{-3}$ in PM2.5) (Zhang et al., 2013) and Delhi, India (16.7 µg m$^{-3}$ in PM10) (Chelani et al., 2010). For

the seasonality, $SO_4^{2-}$ showed equivalent higher concentrations during winter (11.1±11.6 μg m$^{-3}$) and pre-monsoon (10.9±3.33 μg m$^{-3}$) but lower in post-monsoon (6.39±3.90 μg m$^{-3}$). The minimum concentration occurred in monsoon (4.90±3.01 μg m$^{-3}$). It was inconsistent with the seasonal variations of OC and EC (post-monsoon>pre-monsoon>winter>monsoon). However, it is difficult to figure out whether $SO_4^{2-}$ comes from regional transport or local emission, based on our current dataset and discussion. So, such information was not included in our manuscript.

2. In the paragraph starting at Line 247, OC/EC ratio is discussed. The authors need to be more cautious in comparing the OC/EC ratio from their TSP samples to those in the literature, which are largely data associated with PM2.5 samples. OC on the coarse particles has significant contributions from non-combustion sources (such as vegetative detritus, dust). It has been reported OC/EC ratio in the coarse mode was much larger than those in the accumulation mode (e.g. Yu et at, ACP, 10, 5107–5119, 2010), due to the significant presence of non-combustion OC. It would be misleading without commenting on the influence of non-combustion primary OC on the OC/EC ratio.

**Response:** Thank you for pointing this out. Now we marked the particle size of the samples (i.e. PM2.5 and TSP ) in the text (Line 253 and Line 255). We also added some discussion about the uncertainty of OC/EC ratios in TSP samples. "It should be noted that OC may also originate from non-combustion sources such as vegetative detritus and fungal spores in the coarse mode, leading to a high OC/EC ratio in the TSP samples (Yu et al., 2010). Therefore, those bio-aerosols and dust may interfere the OC/EC ratio somewhat and deserve a further study." Please see the details in Line 261 to 264.

3. In the paragraph starting at Line 324, the authors discussed the variation range of Lev/Man ratio and attributed the wide range observed (0.42-22.0) to photochemical degradation of levoglucosan. Such a reason is unlikely, as mannosan also degrades and its degradation rate is likely similar to that of levoglucosan, considering their similar chemical structures.

**Response:** Thank you for your comments. Yes, there is no solid evidence to support the speculation that levoglucosan and mannosan have different degradation rate, especially when we consider they have similar chemical structures. We agree with you that the current statements may be misleading and confusing. Now the corresponding discussion was changed as "The possible reason is due to the different biomass types burning such as the burning of softwood and hardwood. Nevertheless, the mechanism is unclear yet in Lumbini and more detailed studies are needed in the

future." in Line 329 to 332.

4. Lines 130-131: "The aerosol loading is very high at Lumbini,.." Please provide more quantitative information (e.g., annual average or typical seasonal average concentrations).

**Response:** Now we provided more quantitative information about the aerosol loading of Lumbini in Line 132 to 134 as "A previous study reported that daily average PM2.5 (ranging from 6.1 to 272 μg m$^{-3}$ with the average of 53.1±35.1 μg m$^{-3}$) and PM10 (ranging from 10.5 to 604 μg m$^{-3}$ with the average of 129±91.9 μg m$^{-3}$) levels frequently exceeded the WHO guideline value (25 and 50 μg m$^{-3}$ for daily PM2.5 and PM10, respectively) during pre-monsoon season (Rupakheti et al., 2016)."

5. Line 143: The sampling schedule is on a weekly basis. Was it a regular schedule (i.e., one sample every 7 days or a random day in a week)?

**Response:** The sampling schedule was on the basis of a random day in a week, without a clear weekend effect. The choice of the sampling day was also determined according to the weather condition, avoiding the direct disturbance of rain events.

6.Line 152: Please provide more information on the field blank samples: how frequently was field blank filters collected?

**Response:** We add the information about the filed blank samples collecting in Line 153. We collected one field blank sample in each month during the whole sampling campaign.

7. Line 180: 121 m$^3$: does this correspond to the air volume passing through the entire filter? Why it was not 144 m$^3$ (0.100 m$^3$/min $\times$60$\times$24)?

**Response:** Thank you for pointing this out. Actually, the sampling duration time was not exactly fixed in the field, mostly due to the power problem. In most cases, it ranged from 18 to 22 hours. Therefore, we used the average sampling duration (around 20 hours) to estimate the method detection limits. Now we added the explanation in the main text for this point.

**References**

Chelani, A. B., Gajghate, D. G., ChalapatiRao, C. V., and Devotta, S.: Particle Size Distribution in Ambient Air of Delhi and Its Statistical Analysis, Bulletin of Environmental Contamination and Toxicology, 85, 22-27,

10.1007/s00128-010-0010-4, 2010.

Deshmukh, D. K., Tsai, Y. I., Deb, M. K., and Mkoma, S. L.: Characterization of Dicarboxylates and Inorganic Ions in Urban PM10 Aerosols in the Eastern Central India, Aerosol and Air Quality Research, 12, 592-607, 10.4209/aaqr.2011.10.0160, 2012.

Satsangi, A., Pachauri, T., Singla, V., Lakhani, A., and Kumari, K. M.: Water Soluble Ionic Species in Atmospheric Aerosols: Concentrations and Sources at Agra in the Indo-Gangetic Plain (IGP), Aerosol and Air Quality Research, 13, 1877-1889, 10.4209/aaqr.2012.08.0227, 2013.

Zhang, R., Jing, J., Tao, J., Hsu, S. C., Wang, G., Cao, J., Lee, C. S. L., Zhu, L., Chen, Z., Zhao, Y., and Shen, Z.: Chemical characterization and source apportionment of PM2.5 in Beijing: seasonal perspective, Atmospheric Chemistry and Physics, 13, 7053-7074, 10.5194/acp-13-7053-2013, 2013.

Rupakheti, D., Adhikary, B., Praveen, P. S., Rupakheti, M., Kang, S., Mahata, K. S., Naja, M., Zhang, Q., Panday, A. K., and Lawrence, M. G.: Pre-monsoon air quality over Lumbini, a world heritage site along the Himalayan foothills, Atmospheric Chemistry and Physics Discussion, 2016, 1-46, 10.5194/acp-2016-430, 2016.

---

## Author Comment (AC2) · 28 Apr 2017

**Response to referee #2**

We thank the referee for the recognition on our work and the constructive review.

**General comments:**

Wan et al reports a study of organic aerosols in Lumbini, a site of northern Indo-Gangetic Plain (IGP) with the emphases on the influence of biomass burning. They found that organic aerosols in Lumbini was not only related to the local agricultural activities and residential biomass usage, but also impacted by the regional emission. Although some previous work has reported organic aerosols in this region (mostly in India), this study does add valuable knowledge for the scientific community to better understand the abundance, composition and sources of organic aerosols in north IGP. Such information is useful for adapting proper mitigation measures, and thereby reducing the air pollution in South Asia. Generally, this paper is well written and logically ordered. The statement is convincing since it is based on a comprehensive dataset and appropriate discussion. Therefore, I recommend this paper to be published after a revision.

**Specific comments:**

1.The English writing needs to be improved. Some sentences and description are redundant. For example, the sentence of Lines 53-58 is too long, which is hard to follow.

**Response:** Now we carefully checked the English writing. The changes have been marked in the text using red color.

2. Line 177, do you have any idea about the low recovery of malic acid?

**Response:** The exact reason for the low recovery of $D_3$-malic acid is not clear. One possible reason is the loss when spiking the $D_3$-malic acid on the pre-combusted quartz filters. The other potential reason may be due to the low extraction efficiency of $D_3$-malic acid through the mixed of DCM:MeOH (v:v=2:1), which needs to do a further research in the future. Similarly, the low recovery of malic acid (69±6.3%) was also reported by other group using the similar method (Fu et al., 2009).

3. Line 190. K$^+$ should be replaced by major ions, because in the following section, Ca$^{2+}$ and Mg$^{2+}$ were also included in the IC analysis.

**Response:** Changed according to your suggestion. Please see Line 191.

4. Line 200, as stated by the authors, the concentrations of OC, EC and major ions reported in this work were corrected by their field blank values. So, the levels of such constituents in the field blank filters should be presented.

**Response:** Thank you for pointing out. Now we added the field blank values of OC, EC, K$^+$, Mg$^{2+}$ and Ca$^{2+}$ in Line 203 as "which was 0.40 μg m$^{-3}$, 0.01 μg m$^{-3}$, 0.04 μg m$^{-3}$, 0.02 μg m$^{-3}$ and 0.37 μg m$^{-3}$, respectively".

5. Lines 248-259, in general, relatively high OC/EC ratio represents a high contribution of secondary organic aerosol (SOC) as mentioned in many publications. According to your current dataset of OC, EC as well as the tracers, is it possible to estimate the importance of secondary organic aerosols? Considering the emissions and meteorological conditions in this region, SOC maybe also very important, but now little is discussed inside the manuscript.

**Response:** It is difficult to estimate the SOC based on the present organic molecular tracers we analyzed. We agree that SOC may also be important in Lumbini aerosols. We roughly estimate the SOC based on the primary OC/EC ratio (EC-tracer method) (OCpri = EC × (OC/EC)min, OCsoc = OCtot − OCpri) (Turpin and Huntzicker, 1995). According to the EC-tracer method, the annual average concentration of SOC in Lumbini was 14.5±14.0 μg m$^{-3}$, accounting for 37.2% ±20.0% of OC. Please see in Line 420 to 423 with "According to the EC-tracer method (OCpri = EC × (OC/EC)min, OCsoc = OCtot − OCpri) (Turpin and Huntzicker, 1995), we roughly calculated that the annual average concentration of SOC was 14.5±14.0 μg m$^{-3}$, accounting for 37.2% ±20.0% of OC in Lumbini aerosols ".

6. Line 470, due to the low PBL height in the winter, could you provide its average value, instead of the current subjective description?

**Response:** We enhanced the description about the PBL height (with an average of 267.8±63.2 m) in winter and provided the average value in Line 478.

7. Line 505, clear or clean?

**Response:** We changed the "clear" to "clean".

8. I suggested the authors to move the Fig S6, regarding the MODIS image of the regional emission and transport of smoke plume in the fall, to the main text.

**Response:** Thanks for your suggestion. Now we moved the Fig. S6 to the main text as Fig. 10.

9. As highlighted by the authors, the BrC in the organic aerosols deserve more research in the future. Actually, in the paper by Stockwell et al (2016, ACP), some optical properties of BrC in biomass-burning aerosols have already been determined. So, such new progress should be reflected in this manuscript.

**Response:** Now the work of Stockwell et al. (2016) and Pokhrel et al. (2017) were cited in Line 523 and Line 524 to 525.

10. In Figure 5: Fu et al. (2010) was missed in the Section of References.

**Response:** We added the missed reference in Line 581 to 583 in the Section of References.

**References:**

Fu, P., Kawamura, K., Chen, J., and Barrie, L. A.: Isoprene, Monoterpene, and Sesquiterpene Oxidation Products in the High Arctic Aerosols during Late Winter to Early Summer, Environmental Science & Technology, 43, 4022-4028, 10.1021/es803669a, 2009.

Pokhrel, R. P., Beamesderfer, E. R., Wagner, N. L., Langridge, J. M., Lack, D. A., Jayarathne, T., Stone, E. A., Stockwell, C. E., Yokelson, R. J., and Murphy, S. M.: Relative importance of black carbon, brown carbon, and absorption enhancement from clear coatings in biomass burning emissions, Atmos. Chem. Phys., 17, 5063-5078, 10.5194/acp-17-5063-2017, 2017.

Stockwell, C. E., Christian, T. J., Goetz, J. D., Jayarathne, T., Bhave, P. V., Praveen, P. S., Adhikari, S., Maharjan, R., DeCarlo, P. F., Stone, E. A., Saikawa, E., Blake, D. R., Simpson, I. J., Yokelson, R. J., and Panday, A. K.: Nepal Ambient Monitoring and Source Testing Experiment (NAMaSTE): emissions of trace gases and light-absorbing carbon from wood and dung cooking fires, garbage and crop residue burning, brick kilns, and other sources, Atmospheric Chemistry and Physics, 16, 11043-11081, 10.5194/acp-16-11043-2016, 2016.

Turpin, B. J., and Huntzicker, J. J.: Identification of secondary organic aerosol episodes and quantitation of primary and secondary organic aerosol concentrations during SCAQS, Atmospheric Environment, 29, 3527-3544, 10.1016/1352-2310(94)00276-q, 1995.

---

## Author Response (AR2)

**Response to Co-Editor's comments**

We appreciate the detailed and constructive comments and suggestions from you for. The point-by-point answers to the comments were listed as below.

**Major comment:**

In responding to reviewer comments, results of the EC-tracer method have been added related to the contribution of secondary sources to organic aerosol. The addition is very brief and is not rigorously discussed, unlike the other results presented in the paper. Thus, I ask the authors to develop these results further by separating it from the section dedicated to biomass burning contributions to OC and providing a discussion of these results in relation to prior studies in the region (as has been done for biomass burning).

**Detailed/editorial comments are:**

1. It is not clear if the IPCC estimates of radiative forcing include indirect effects (or not), please clarify this in the first paragraph of the introduction.

**Respond:** According to the report of IPCC 2013, it was stated as "the radiative forcing was due to aerosol-radiation interactions (RFari), which encompasses radiative effects from anthropogenic aerosols before any adjustment takes place, and corresponds to what is usually referred to as the aerosol direct effect. Additional RFari contributions from biomass burning emissions was +0.0 (–0.2 to +0.2) W m$^{-2}$, which here the RFari is made up of a positive black carbon RFari and a negative organic carbon.". Here, we clarify the expression as "The assessment reports by the Intergovernmental Panel on Climate Change (IPCC) have reported that the **direct radiative forcing** of BC and OC from the biomass burning emissions can offset each other to give an estimated **direct RF** of +0.0 (–0.2 to +0.2) W m$^{-2}$" in line

53-56.

2. Line 179 should read "but no target compounds were detected."

**Respond:** Changed as you suggested in line 179.

3. Line 182 should read "for an average total standard volume of 121 m$^3$." (Note: the units are cubic meters, not inverse cubic meters.)

**Respond:** Changed as you suggested in line 182.

4. Line 185 "gradient" should be deleted. Instead, "HPLC-grade dicholoromethane and methanol…"

**Respond:** Changed as you suggested in line 185.

5. Line 203 should read "which were…"

**Respond:** Changed as you suggested.

6. In section 3.1 please clarify the meaning of the errors associated with the average values – are these the standard deviation?

**Respond:** Yes, those errors are the standard deviation. Now we clarified that with (arithmetic mean ± standard deviation) in line 228.

7. Line 250 should include one standard deviation with the average OC/EC ratio.

**Respond:** We added the deviation (±2.09) of the average OC/EC ratio in line 251.

8. At the beginning of the discussion the OC/EC data, please restate that the measurements in this study are TSP.

**Respond:** We changed the expression as "The OC/EC ratios in TSP of our study were relatively high" in line 250.

9. Line 263, "bioaerosols" without the hyphen.

**Respond:** We changed the "bio-aerosols" to "bioaerosols" in line 265.

10. At line 420, the results of the EC tracer method should be presented in a different subsection (either a new section numbered 3.4 or with the ECOC data.). It does not fit within the "biomass burning contribution to OC" part, as it focuses on secondary sources.

**Respond:** Yes, we agree. Now we add a new section 3.4 regarding the SOC. More details could be found in the reply of comment 11.

11. Further, results of the EC tracer method need to be discussed in relation to prior studies in the region. It could also be more robust if correlation with secondary inorganic aerosol (e.g., sulfate) were performed and discussed.

**Respond:** We discussed SOC based on the EC-tracer as "In addition to the contribution of primary biomass burning to OC, SOA formation from biomass burning (BB-SOA) should also be taken into consideration, since BB-SOA such as some phenol compounds and methoxyphenols can account for a significant fraction of particulate matter derived from wood combustion, accounting for 21% and 45% by total aerosol masses (Hawthorne et al., 1989; Yee et al., 2013; Zhang et al., 2015).

EC can be considered as a good tracer of primary combustion-generated carbonaceous aerosols. Therefore, according to the EC-tracer method (OCpri = EC × (OC/EC)min, OCsoc = OCtot − OCpri) (Turpin and Huntzicker, 1995), we roughly calculated that the annual average concentration of secondary organic carbon (SOC) was 14.5±14.0 μg m$^{-3}$, accounting for 37.2%±20.0% of OC in Lumbini aerosols. Obviously, it was also a major contributor to OC. Compared with the previous

studies, the averaged SOC based on the EC-tracer in suburban Dayalbagh, IGP was $13.2\pm10.8$ µg m$^{-3}$ (Satsangi et al., 2012), which was similar with our study, but much higher contribution to OC with 49.0%−55.0%. Using the same method, Shakya et al. (2010) estimated 31% of the SOC to OC in the urban area of Kathmandu during the wintertime due to the biomass burning influences. Ram and Sarin, (2010) also evaluated ~30% of SOC to OC at sampling locations in northern India during wintertime, attributing to the relative dominance of OC derived from wood-fuel and agriculture-waste. Additionally, a close relationship was observed between BB-OC and SOC (Fig. S6a, $R^2$=0.40, P<0.001) but not between $SO_4^{2-}$ and SOC (Fig. S6b, $R^2$=0.10, P<0.05), indicating the predominant role of BB-derived volatile organic compounds in SOC formation in Lumbini." in line 419 to 438.

[Figure]

**Fig. S6.** Correlations between SOC and BB-OC, SOC and $SO_4^{2-}$ during different seasons in Lumbini aerosols.

12. In Table 1, please add either to the table or the caption which months are included in the "pre-monsoon, monsoon, post-monsoon, and winter" sections.

**Respond:** We added the months of each season in the caption of Table 1 in line 834-835.

13.  In Figure 8, please leave a gap between missing data (e.g. May) rather than when connecting with a line.

**Respond:** We changed the Figure according to your suggestion.

**References**

[revised manuscript text omitted]